# Finite-Element Simulation for Thermal Modeling of a Cell in an Adiabatic Calorimeter

**José Eli Eduardo González-Durán** [1,†], **Juvenal Rodríguez-Reséndiz** [2,*,†],
**Juan Manuel Olivares Ramirez** [3,†], **Marco Antonio Zamora-Antuñano** [4,†]
**and Leonel Lira-Cortes** [5,†]

1   Instituto Tecnológico Superior del Sur de Guanajuato, Guanajuato 38980, Mexico; je.gonzalez@itsur.edu.mx
2   Facultad de Ingeniería, Universidad Autónoma de Querétaro, Querétaro 76010, Mexico
3   Universidad Tecnológica de San Juan del Río, San Juan del Río 76800, Mexico; jmolivar01@yahoo.com
4   Departamento de Ingeniería, Universidad del Valle de Mexico, Querétaro 76230, Mexico;
    marco.zamora@uvmnet.edu
5   Centro Nacional de Metrología, El Marques 76246, Mexico; llira@cenam.mx
*   Correspondence: juvenal@uaq.edu.mx; Tel.: +52-442-192-1200
†   These authors contributed equally to this work.

**Abstract:** This research obtains a mathematical formulation to determine the heat transfer in a transient state, in a calorimeter cell, considering an adiabatic system. The development of the cell was established and the mathematical model was transiently solved, which approximated the physical phenomenon under the cell operation. A numerical method for complex geometries was used to validate performance. The results obtained in the transient heat transfer in a cylinder under boundary and initial conditions were compared using an analytical solution and numerical analysis employing the finite-element method with commercial software. The study from the temperature distribution can afford, selection between a cylindrical and spherical geometry, design criteria that are generated by changing parameters such as dimension, temperature, and working fluids to develop an adiabatic calorimeter to measure the heat capacity in fluids. We show the mathematical solution with its initial and boundary conditions as well as a comparison with a numerical solution for a cylindrical cell with a maximum error from 0.075% in the temperature value, along with a theoretical and numerical analysis for a temperature difference of 1 °C.

**Keywords:** adiabatic calorimetry; numerical simulation; heat capacity; finite-element method; heat transfer

## 1. Introduction

For engineering design, with heat exchange in, for example, refrigerators, the car engine, the food industry, and the heat treatment of materials, the research or development of new substances as refrigerants or working fluids is necessary. Among these substances is nanofluid, which has been recently introduced and improves heat transfer in geometries such as enclosures, channels, and microchannels [1]. For the engineering applications mentioned above, it is necessary to know the thermophysical properties of the working fluids that are involved in the energy exchange processes. One way to know how efficient the working fluid is for the exchange of energy in the form of heat is through thermal efficiency. However, to calculate the thermal efficiency, it is necessary to know the value of the heat capacity of the working fluid. The amount of heat required to increase the temperature by one degree of a substance is known as heat capacity. Therefore, it is important to measure the heat capacity as accurately as possible for the efficient use and development of the fluids involved in the exchange of heat. In [2], heat capacity is an important variable to calculate the thermal

efficiency in any system that involves heat transfer. The most reliable way to obtain the value of heat capacity is by an adiabatic calorimetry technique [3]. Certain researchers have developed their own devices that work under this principle [4–9] with different configurations for the geometry of the cell in their adiabatic calorimeters. The cell is a key element of the calorimeter containing the fluid to measure its heat capacity. It has other elements such as sensors, heaters, inlets, and outlets for the fluid. According to adiabatic calorimetry, the cell is surrounded by a shield called an adiabatic, and its function is to stay at the same temperature of the cell to avoid heat transfer between both [9]. The cell and its elements are located within a cryostat, which is responsible for generating an environment with a constant temperature. The working principle is as follows: A quantity of test fluid is set inside of a cell to evaluate the heat capacity. When the fluid is inside a cell, heat is added and the temperature of the fluid consequently increases. The amount of heat ($Q$) added, the mass of fluid ($m$) contained in the cell, and the variation of the temperature ($T$) of fluid are known [9]. It is possible to calculate heat capacity from the next equation:

$$Q = (mC_p)\Delta T \tag{1}$$

where $C_p$ is the heat capacity of the sample cell, and the heat added is obtained by the electrical energy calculated by

$$Q = IVt \tag{2}$$

where $I$, $V$, and $t$ are the current, voltage, and duration of heating, respectively [7], taking into account their respective uncertainties. In the laboratory of thermophysical properties of Centro Nacional de Metrología (CENAM), they are developing an adiabatic calorimeter for measuring the heat capacity of fluids at room conditions. One of the most critical elements of the adiabatic calorimeter is the cell, which contains the fluids to determine the value of the heat capacity. Then to developing the adiabatic calorimeter, it is necessary to start to analyze the temperature distribution in the cell for the thermal design criteria regarding the cell; in the ideal case, the temperature of fluid contained in the cell is constant or equal to any point, but in a real case, temperature gradients are present. Therefore, mathematical models were used to analyze the temperature distribution in the cell from this work, as in other works, e.g., [10], which focused on finding errors in the thermal conductivity measurement process, [11], which studied the impact of the geometry for heat transfer in nanofluids, and [12], which studied a Spray Fluidized Bed Granulator. However, from the literature, no works related to the analysis of the thermal performance to select the geometry of a cell in an adiabatic calorimeter have been found. Attempts to learn the temperature distribution within the cell, the location of the temperature sensor, and the fluid inlets and outlets can be studied by inverse heat conduction problem [13].

Numerical analysis was used because it is a tool to change parameters such as dimensions, geometric shapes, and initial and boundary conditions to analyze the behavior of a mathematical or physical model in less time. However, the mathematical and numerical models provide only approximations to the physical phenomenon, so it is essential to validate the results, and the best way is to use a prototype and to characterize it [12,14–16].

The use of simulation is a fundamental aspect of developing an engineering process, research for scientific development, and the characterization of geometrical parameters, such as in [17]. To verify the accuracy of numerical methodologies, numerical results are compared with analytical solutions [17–19]. In other research, three-dimensional governing equations (continuity, momentum, and energy) along with boundary conditions are solved using the finite volume method and other processes of scientific applications in the study of energies.

Therefore, the following formulations were made: theoretical and numerical analysis by the finite-element method for a cylindrical cell and only a numerical analysis for a spherical cell. The present work performs a comparison among the results of the analytical and numerical solutions

for a cylindrical cell, and the results of numerically evaluating a spherical cell are also included. Moreover, the temperature distribution between spherical and cylindrical cells, as well as the effect of varying their dimensions, is compared and evaluated.

The main objective of this work was to select the geometry with the lowest temperature gradients through carrying out an analytical solution of the thermal behavior of the cell from an adiabatic calorimeter to later compare it with a numerical solution, and to determine the error between both methods to provide greater reliability in the simulation of processes, where there is an exchange of energy with geometries of greater complexity, allowing for a smaller number of experimental prototypes that will reduce time and cost in the design of any thermal system. In addition, the calculated information given by the simulation contributes to a criteria design. It permits to choose a spherical geometry instead of a cylindrical, as reported in the state-of-the-art.

This article is organized as follows: Section 1 describes the adiabatic calorimeter, and works reported in the literature on the working principle of this type of device are reported. Section 2 presents the mathematical formulation wherein we establish a partial differential equation in a transient state, which was solved step by step in an analytical way with Bessel functions and their eigenfunctions. The solution was programmed in a Matlab language to compare results with the finite-element method. Section 3 shows the numerical model used to simulate the operation of the cell and the parameters used in the numerical model, and its boundary and initial conditions such as the element type chosen for the solution by ANSYS are also shown. Section 4 depicts a comparison of the results obtained through theoretical analysis and the numerical model used to show the power of using the numerical model to solve this kind of device and process as well as how useful it is to approximate physical phenomena. Section 5 gives an analysis of the numerical tools and its accuracy concerning an analytical model. Finally, Section 6 shows the conclusions based on theoretical and numerical results analysis done with regard to a specific part of an adiabatic calorimeter.

## 2. Theoretical Analysis

In this section, we show the mathematical model in a transient state established to describe the behavior of a cell from an adiabatic calorimeter, it was simplified by its symmetry along axis $z$. Several subsections were added, and they include the solution for temporal and spacial coordinates and the application of boundary conditions to find the eigenvalues. The mathematical model was solved step by step in an analytical way, and the final solution was programmed in Matlab to compare the results obtained with the finite-element method in the following section.

*Mathematical Formulation*

The equation of heat conduction is set in a cylindrical coordinate system for a theoretical analysis because the proposal cell presents this geometry [13].

The differential equation of heat conduction in the cylindrical coordinate system is [20]:

$$\frac{\partial^2 T}{\partial r^2} + \frac{1}{r}\frac{\partial T}{\partial r} + \frac{1}{r^2}\frac{\partial^2 T}{\partial \phi^2} + \frac{\partial^2 T}{\partial z^2} = \frac{1}{\alpha}\frac{\partial T}{\partial t} \tag{3}$$

where

$$\alpha = \frac{k}{\rho c_p}$$

where $\alpha$ is the thermal diffusivity, $k$ is the thermal conductivity, $\rho$ is the density, and $c_p$ is the heat capacity of the material to be analyzed. The cell is a solid of radius $b$ and height $c$, and its boundaries are a temperature constant $T_c$ [3,5,6,8,21,21,22]. Initially, the entire system is at temperature $T_0$, where there is no heat generation inside the cell, as Figure 1 shows together with the coordinate system. Under these conditions and because there is azimuthal symmetry, $T$ is independent of $\phi$, so the equation for the solution is

$$\frac{\partial^2 T}{\partial r^2} + \frac{1}{r}\frac{\partial T}{\partial r} + \frac{\partial^2 T}{\partial z^2} = \frac{1}{\alpha}\frac{\partial T}{\partial t} \tag{4}$$

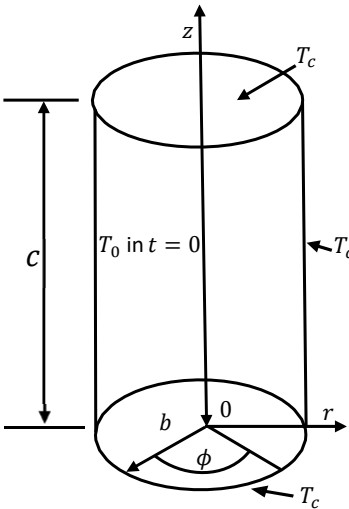

**Figure 1.** The cylinder with a radius and height specified, under certain initial and boundary conditions in the cylindrical coordinate system.

The initial condition is at $t = 0$ with $T = T_0$ at $0 \leq r \geq b$; $0 \leq z \geq c$ and $t > 0$, a boundary condition $T = T_c$, at $r = 0$, $r = b$, $z = 0$, and $z = c$. This assumes that the variables are separated $T(r, z, t) = R(r)Z(z)\Gamma(t)$. Substituting in Equation (4), simplifying, and dividing by $RZ\Gamma$,

$$\frac{1}{R}\left(\frac{d^2 R}{dr^2} + \frac{1}{r}\frac{dR}{dr}\right) + \frac{1}{Z}\frac{d^2 Z}{dz^2} = \frac{1}{\alpha\Gamma}\frac{d\Gamma}{dt} \tag{5}$$

Thus, in Equation (5), the left side is spatial coordinates, and the right side is time coordinates. The differential equation is equal to a constant

$$\frac{1}{\alpha\Gamma}\frac{d\Gamma}{dt} = -\lambda^2 \tag{6}$$

Therefore, a temporary solution is

$$\Gamma(t) = \Gamma(0)e^{\alpha\lambda^2 t} \tag{7}$$

For the axis $z$,

$$\frac{1}{Z}\frac{d^2 Z}{dz^2} = -\eta^2 \tag{8}$$

$$\frac{d^2 Z}{dz^2} + Z\eta^2 = 0 \tag{9}$$

and the solution to Equation (9) is

$$Z = A\cos(z\eta) + B\sin(z\eta) \tag{10}$$

where grouping constants $A = (A + B)$ and $B = (iA + iB)$

Solving the spatial part for the axis $r$, we have

$$\frac{1}{R}\left(\frac{d^2R}{dr^2} + \frac{1}{r}\frac{dR}{dr}\right) = -\lambda^2 + \eta^2 = -\beta^2 \tag{11}$$

and Equation (11) is a differential equation from Bessel of the order $v$ and is written as

$$\frac{d^2R_v}{dr^2} + \frac{1}{r}\frac{dR_v}{dr} + \beta^2 R_v = 0 \tag{12}$$

The solutions to Equation (12) are the Bessel functions of order $v$, which are

$$R_v(\beta r) = \{J_v(\beta r), Y_v(\beta r)\} \tag{13}$$

With the superposition of all solutions, Equation (5) is

$$T(r,z,t) = \sum_{m=1}^{\infty}\sum_{p=1}^{\infty} R_v(\beta_m r) Z(\eta_p, z)\Gamma(t) \tag{14}$$

where $R_v(\beta_m r)$ and $Z(\eta_p, z)$ are the eigenfunctions, the solutions of the equations separated; $\beta_m$ and $\eta_p$ are the respective eigenvalues.

To find the value of $\beta$, a change of variable is made, and $T^* = T - T_c$ is defined from

$$R_v(\beta r) = AJ_v(\beta r) + BY_v(\beta r) \tag{15}$$

with the boundary condition at $r = 0 \rightarrow T = T_c \Rightarrow T^* + T_c = T_c \rightarrow T^* = 0$, substituting into Equation (15). Since $r = 0$, $R_v(\beta r)$ is finite:

$$BY_v(\beta r) = 0 \Rightarrow B = 0$$

From the boundary condition at $r = 0$,

$$AJ_v(\beta b) = 0 \tag{16}$$

$A$ cannot be 0, therefore the values of $\beta$ are the positive roots of

$$J_v(\beta_m b) = 0 \tag{17}$$

Since the problem includes the origin, $Y_v(\beta r)$ implies that $v = 0$, which is divergent.

To find $\eta$, the equation

$$Z = A\cos(z\eta) + B\sin(z\eta) \tag{18}$$

and the boundary condition at $Z(z = 0) = T^* = 0$ implies that $A = 0$. The boundary condition at $Z(z = c) = T^* = 0$ entails

$$B\sin(c\eta) = 0 \tag{19}$$

As a result of Equation (19), $B$ cannot be 0. The values of $\eta$ are

$$\eta_p = \frac{p\pi}{c} \tag{20}$$

The solution of the equation is

$$T^*(r,z,t) = \sum_{m=1}^{\infty}\sum_{p=1}^{\infty} C_m J_0(\beta_m r)\sin(\eta_p z)e^{-\alpha\beta^2 t} \tag{21}$$

Applying the initial condition to Equation (21), we have

$$T^*(r,z,t=0) = T_0 - T_c = \sum_{m=1}^{\infty} \sum_{p=1}^{\infty} C_m J_0(\beta_m r) \sin(\eta_p z) \tag{22}$$

To find the constant $C_m$,

$$C_m = \frac{1}{N(\beta_m)N(\eta_p)} = \int_0^b \int_0^c r J_0(\beta_m r') \sin(\eta_p z') dr' dz' \tag{23}$$

To find the norms, for $N(\beta_m)$,

$$N(\beta_m) = \int_0^b r J_0^2(\beta_m r) \tag{24}$$

From Özisik [20], we have

$$N(\beta_m) = \frac{b^2}{2} \left[ J_v'^2(\beta_m b) + \left( 1 - \frac{v}{\beta_m^2 b^2} \right) J_v^2(\beta_m b) \right] \tag{25}$$

With $v = 0$,

$$N(\beta_m) = b^2 [J_0'^2(\beta_m b)] \tag{26}$$

To find $N(\eta_p)$,

$$N(\eta_p) = \int_0^c \sin(\eta_p z) \sin(\eta_p z) dz \tag{27}$$

Solving the previous equation, we have

$$N(\eta_p) = \frac{c}{2} \tag{28}$$

Substituting in Equation (14), we have

$$T^*(r,z,t) = \sum_{m=1}^{\infty} \sum_{p=1}^{\infty} \frac{1}{N(\beta_m)N(\eta_p)} J_0(\beta_m r) \sin(\eta_p z) e^{-\alpha\beta^2 t} \int_0^b \int_0^c r J_0(\beta_m r') \sin(\eta_p z') dr' dz' \tag{29}$$

Finally, the solution according to the initial and boundary conditions posed is

$$T^*(r,z,t) = \sum_{m=1}^{\infty} \sum_{p=1}^{\infty} \frac{J_0(\beta_m r) \sin(\frac{p\pi}{c} z) e^{-\alpha[\beta_m^2 + (\frac{p\pi}{c})^2]t}}{\frac{1}{2} c b^2 [J_0'^2(\beta_m b)]}$$
$$(T_0 - T_c) \left[ \frac{b}{\beta_m} J_1(\beta_m b) \right] \left[ -\frac{c}{p\pi}(-1)^p + \frac{c}{p\pi} \right] + T_c \tag{30}$$

Equation (30) was programmed in Matlab, and the input data required are: the initial temperature $T_0$, the temperature at the border $T_c$, the thermal diffusivity value $\alpha$ is according to the fluid to be analyzed, the radius $r$, the height $c$, and the number of divisions required in the discretization, the initial and final time and the time steps. All that data are used to obtain the temperature in coordinates $r, z$ and any time $t$. In the solution obtained convergence is achieved with 65 values for $p$ and for $m$ were set to 200. The finite-element method was used with commercial software to compare the results obtained from the analytical solution [1,23].

## 3. Numerical Solution

This section describes the numerical model used for comparison with the analytical solution, and all parameters such as boundary and initial conditions, and the dimensions of the cylinder were used the same as for analytical solution.The numerical solution was solved via the finite-element method by commercial software.

*Model to Compare the Analytical Solution*

We used the ANSYS Mechanical Parametric Design Language (APDL) user interface 18.0, which uses the finite-element method to approximate the solution of transient heat transfer conduction by Equation (4) shown above [10,23] for the numerical solution, running in a workstation with a Xeon processor with 16 cores and 16 GB of RAM.

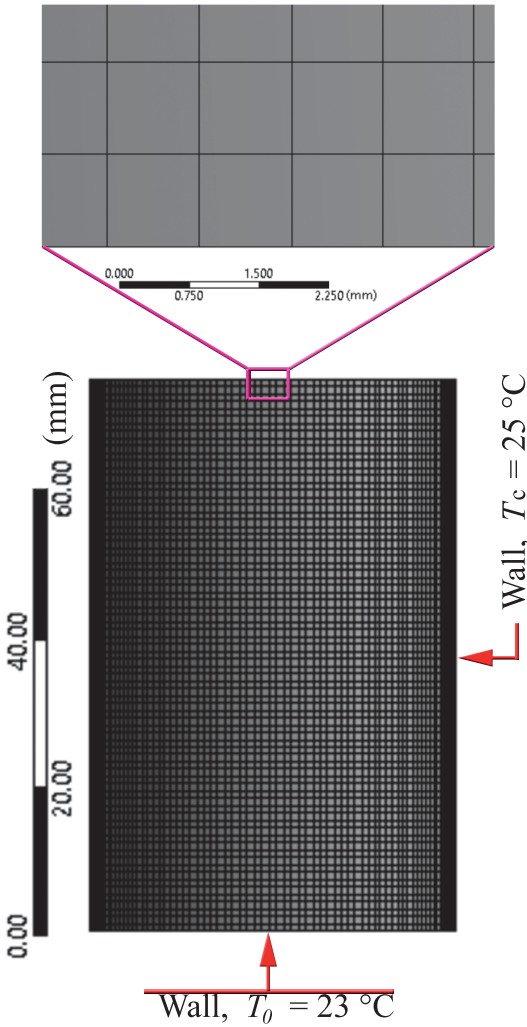

**Figure 2.** Mesh for the numerical model.

The model used in commercial software is two-dimensional and axisymmetric, which allows for the condition of azimuthal symmetry established for the analytical problem. The type of element that was used is a plane 77, which is an 8-node element for thermal analysis that accepts axisymmetry conditions and transient or steady-state analysis. Taking into account the numerical solutions, the grid number was investigated as a parameter that affects the accuracy of the obtained results. Based on the variations of these parameters in a grid number of 5610 nodes, we can be assured that the obtained results are accurate, because several simulations were computed varying mesh size and time step for convergence until get the minimum percentage which was 0.075% error against the theoretical solution. The surface was meshed using a quadrilateral dominant method as displayed in Figure 2. The solution of heat conduction problems is governed by the characteristic time of heat diffusion. The mesh size is less important. To calculate the numerical solution, a cylinder with the following properties was selected: the thermal conductivity essential variable in the model [2], heat capacity, and density of water [24]. The dimension of the cylinder are as follows: radius: 24.5 mm; height:

74.24 mm. The boundary conditions are a constant temperature of $T_c = 25\,°\text{C}$. The initial condition $T_0$ at $t = 0$ was established at $T_0 = 23\,°\text{C}$. The study was solved in a transient mode for a physical time of 6000 s with a time step size of 0.1, for a minimum of 0.05, maximum of 1.5, and a maximum numbers of iterations of 100.

## 4. Analysis Results

This section shows a comparison of the results between the analytical and numerical solutions. Another geometry is included, i.e., the spherical one, and was analyzed only by the numerical method. The results are shown in four different sizes of cylinders with the same sphere of 2.5 cm radius. Where the analytical solution named "CYLINDRICAL" is compared with solutions obtained by the finite-element method, where the cylinder is named "MEF CYLINDRICAL," and the sphere is named "MEF SPHERICAL," representing the point at which the temperature gradients are greatest and are at the centroid of the sphere and the cylinder. To reduce the solution time and the amount of data, a constant of thermal diffusivity value was considered as the unit instead of the water value of $0.14 \times 10^{-6}\ \text{m}^2/\text{s}$.

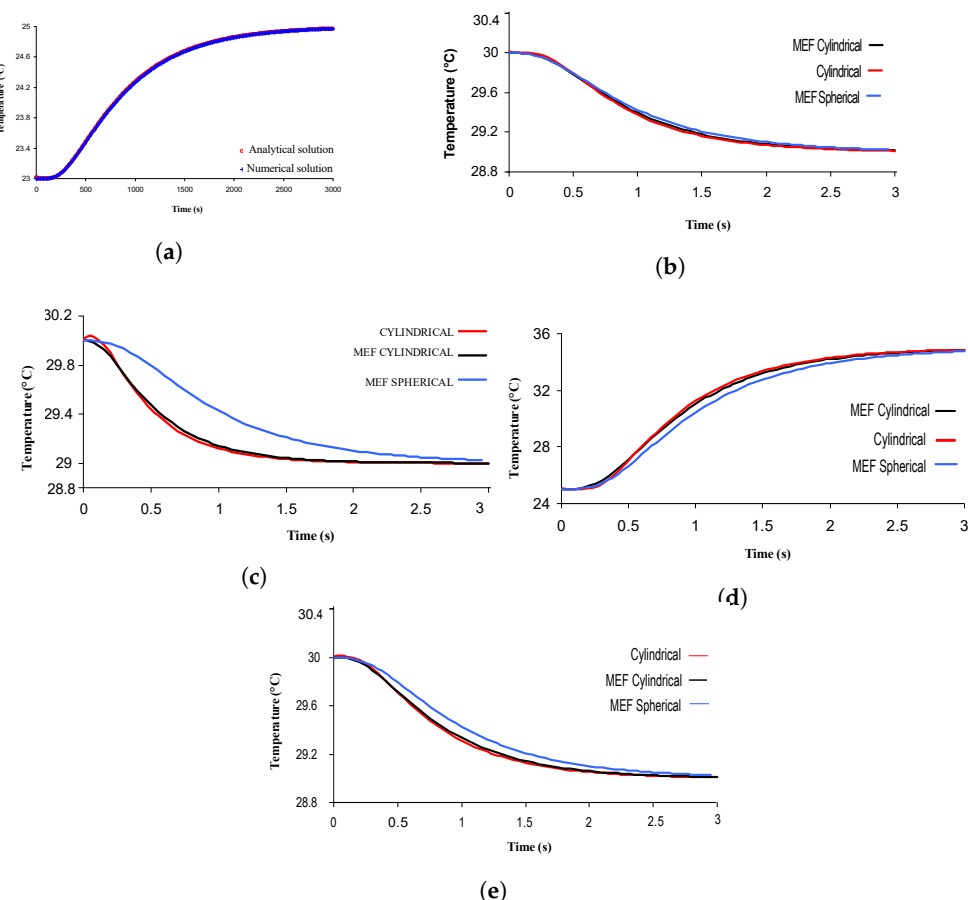

**Figure 3.** Comparison between analytical and numerical solutions for: (**a**) the centroid of a cylinder, located at a radius of 0.0 cm and a height of 3.71 cm with properties of the water; (**b**) a cylinder with a radius of 2.2 cm and a height of 4.5 cm, compared with the sphere with a 2.5 cm radius; (**c**) a cylinder with a radius of 1.5 cm and a height of 9.9 cm, compared with the sphere with a 2.5 cm radius; (**d**) a cylinder with a radius of 2.2 cm and a height of 4.5 cm, compared with the sphere with a 2.5 cm radius; (**e**) the initial condition of 25 °C and the boundary condition of $T_c = 35\,°\text{C}$ for a cylinder with a radius of 2.2 cm and a height of 4.5 cm, compared with the sphere with a 2.5 cm radius.

Figure 3a plots the evolution of temperature according to the boundary conditions mentioned: $T_0 = 23\,°\text{C}$ at $t = 0$ and constant temperature $T_c = 25\,°\text{C}$ for 6000 s with water, such as the fluid inside

the cell. The point analyzed exactly was the centroid of the cylinder, according to the coordinates plotted in Figure 1, the location of centroid is $r = 0$ and $z = 3.712$ cm. The curve depicted in Figure 3a has a maximum error percentage of 0.075% between the analytical and numerical solution with a temperature increase of 2 °C.

Figure 3b,c,e compares temperatures over time with the initial condition of $T_0 = 30$ °C and with the boundary condition of $T_c = 29$ °C for different cylinder dimensions and a sphere with a radius of 2.5 cm. The dimensions of the cylinder with a smaller temperature gradient have a 1.5 radius and a 9.9 cm height. This reached the steady state at the highest speed with respect to the other cells; at a time of 1.5 s, the cylinder reached a uniform temperature compared with the other configurations, which took almost twice as long [24,25].

For the next case, Figure 3d shows a comparison among analytical and numerical solutions for an initial condition of $T_0 = 25$ °C and a boundary condition of $T_c = 35$ °C. The numerical result of the sphere with a radius of 2.5 cm is included. The error obtained is about 0.86%. Among all analytical and numerical solutions, this was the highest, and it is because there is a big difference in temperature, 10 °C, in contrast to the 1 °C difference in the other cases [24,25].

## 5. Discussion

This work demonstrates the accuracy of analytical and numerical solutions in the development of an adiabatic calorimeter. It is easy to avoid experimental considerations during digital measurements. The adiabatic condition cannot be achieved. It is essential in this apparatus to keep constant temperature of the cryostat and the same temperature between cell and its adiabatic shield. Thus, it is possible to calculate heat leaks in this kind of apparatus to determine the heat capacity more accurately as possible.

An accurate measurement of heat capacity is dependent on the temperature rise measurement and the change-of-volume work adjustment [9]. Thus, this work focuses in the gradients temperature generated by the temperature rise. The main result of this work was shared because [9] chose a spherical geometry unlike [3–8]. It was figured out that spherical geometry has lower gradients that cylindrical one.

Matlab is a high-level tool for the development of applications in engineering and research, as in this work. Based on the operation of the adiabatic calorimeter, was simplified the mathematical model that approximates the operation of a cell in an adiabatic calorimeter and it was programmed in Matlab to analyze the variables that affect the exchange of energy. Different values were considered for it, the temperature, time, and fluid properties. Simulation research provides a reference for engineering applications and excellent tools to evaluate physical phenomena taking account of complex geometries.

Numerical simulations using a finite-element method were performed using the package ANSYS 18.0 (ANSYS, Inc., Canonsburg, PA, USA) to understand the behavior of heat transfer and heat leaks. Through a heating resistor, which was conventionally used during the period of operation, the operating conditions used in numerical simulations can be checked. The performance of the calorimeter can be tested. Measurements can be made to assess improvements in sensitivity. There are sources of heat that cannot be accounted for, e.g., temperature sensors and heat losses through the lid decrease as heat generation and thermal fluid speed increases. Taking into account [10,11], the analytical solution developed in this work is at most a 3% difference for experimental results that could be obtained.

The typical procedure for loading a sample into a calorimetric device is as follows: The pre-packed sample container is filled, and the sample is hermetically sealed on the device. The assembly is reweighed to determine the mass of the sample. The sample vessel is placed in the heater/thermometer assembly. The calorimeter temperature increases stepwise during the heat pulse. This is to minimize its contribution to the total heat load. For most of the equilibrium period, the calorimeter temperature is almost entirely constant because the device deliberately turns on and off every specific time interval. Other studies with calorimeters are the enthalpy-temperature ratio in materials have been conducted

by [24,25]. The most recent studies have followed the approach that uses essential thermodynamic relationships [26–28].

The calorimetry allows one to study the thermal behavior in the heat exchange that occurs in a system, managing to understand more clearly the heat transfer mechanism in it, and providing data that helps its characterization in materials and substances. In engineering, its contribution can be extended to different parts of the optimization process, to chemical reactions, and to operating units, i.e., in any process where energy exchange is present. The calorimetry in the food industry, it helps in food preservation by determining the optimal proportions of compounds. It contributes to the study of vegetable crops to analyze germination processes, it aids in the quantification of nanosolids for pharmaceutical use, and it helps in the thermal characterization of nano-structured lipid transporters. This has been made possible in contrast with the works accomplished by [29,30].

## 6. Conclusions

A strategy was established to improve computational time, as described below. The first analysis shown in Figure 3a, was computed using properties of water to simulate this fluid inside of the cell. For the simulations shown in Figure 3b–e, a thermal diffusivity of value 1 was used to reduce computational time. The time is only 3 s, in contrast with Figure 3a, where time is 3000 s.

The maximum error obtained in this work among the analytical results and the numerical method was around 0.86% for an increase of 10 °C and the minimum was 0.075% for an increase of 1 °C. These results show that there is a good agreement with the finite-element method.

The cylinder of a 1.5 cm radius and a height of 9.9 cm reached in a period time of 1.5 s a uniform temperature inside of the entire cell, which is half the time with respect to other dimensions evaluated. This is because the evaluated cylinders have a smaller radius with respect to the sphere. However, other parameters such as temperature sensor location, the cell material, and the inlets and outlets for fluid to the cell need to be evaluated.

The spherical geometry has better thermal performance than the cylindrical, because the temperature gradients are smaller. The results obtained are that the maximum gradient for a spherical cell is 0.104 °C, and the maximum gradient for a cylindrical cell is 0.208 °C; therefore, geometry affects thermal behavior, as reported by [2]. In real applications, the system in a sphere responds with greater thermal speed than in a system contained within a cylinder. However, the homogeneity in the fluid contained in the cell is the most important variable because accuracy measurements affect the heat capacity value.

When heat is added to a system, increasing the temperature from 23 °C to 25 °C, the difference in temperature between the analytical solution and the numerical solution is 0 as seen in Figure 3a. The maximum temperature variation between the cylinder and the sphere obtained from the simulation occurred at the time of extracting heat from the system, causing a decrease in temperature from 30 °C to 29 °C, generating a temperature variation of 0.08 °C according to Figure 3e.

The error among analytical and numerical results increases with increasing temperature and decreases as the steady state is reached.

**Author Contributions:** Conceptualization, J.E.E.G.-D. and J.M.O.R.; methodology, J.E.E.G.-D., J.M.O.R., and M.A.Z.-A.; software, M.A.Z.-A., J.M.O.R., and J.E.E.G.-D.; validation, J.E.E.G.-D., J.R.-R., M.A.Z.-A., J.M.O.R., and L.L.-C.; formal analysis, L.L.-C., J.E.E.G.-D., J.R.-R., and M.A.Z.-A.; writing—original draft preparation, M.A.Z.-A., L.L.-C., and J.R.-R.; writing—review and editing, M.A.Z.-A.; supervision, M.A.Z.-A., J.R.-R., and L.L.-C. All authors have read and agreed to the published version of the manuscript.

**Funding:** This research was funded by the Consejo Nacional de Ciencia y Tecnología (CONACYT) and PRODEP.

**Conflicts of Interest:** The authors declare that there is no conflict of interest.

## Abbreviations

The following abbreviations are used in this manuscript:

| | |
|---|---|
| $Q$ | Amount of heat |
| $m$ | Mass |
| $\Delta T$ | Temperature variation |
| $I$ | Current |
| $V$ | Voltage |
| $t$ | Time |
| $^\circ$C | Celsius grade |
| $\alpha$ | Thermal diffusivity |
| $k$ | Thermal conductivity |
| $\rho$ | Density |
| $C_p$ | Heat capacity |
| $b$ | Constant in radial coordinates |
| $c$ | Constant in $z$ coordinates |
| $T_c$ | Temperature constant |
| $T^*$ | Represent a change of variable for T |
| $T_0$ | Initially temperature |
| $R_v$ | Bessel functions of order $v$ |
| $R_v(\beta_m r)$ | Eigenfunction |
| $Y_v$ | Bessel functions of order $v$ |
| $J_v$ | Solution for Bessel functions of order $v$ |
| $\beta_b$ | Eigenvalue evaluated in radius $b$ |
| $\beta_m b$ | Summation of eigenvalue evaluated in radius $b$ |
| $Z\eta p$ | Eigenfunction for $z$ coordinate |
| $\eta_p$ | Summation of eigenvalue evaluated in height |
| $C_m$ | Normal function |
| $J_0$ | Bessel functions evaluated in 0 |
| CENAM | Centro Nacional de Metrologia |

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
