# Peer review of "Finite-Element Simulation for Thermal Modeling of a Cell in an Adiabatic Calorimeter"

_energies, doi:10.3390/en13092300_

Round 1

Reviewer 1 Report

This is a well-organized manuscript and written in clear language, but following issues should be addressed to improve the content:

  1. In the introduction section, the authors discuss the structure of this manuscript, which is very clear. However, I don’t see much discussion about previous theoretical and numerical studies, and how this work is different from them. The motivation behind this work was not very clear. Thus, I would suggest the authors reframe some portions of the introduction section and include more references to highlight the motivation and novelty of the work.
  2. In the caption of Fig.3, c) and d) seem to be the same; however, the corresponding figures are different. Please revise them. Also, the dimensions of the cylinder or sphere of each case can be added to the figure to make it easier for readers.
  3. Does the numerical model in Fig. 2 shows the one for the cylindrical model? Is it possible to show the spherical model as well?

Author Response

REVIEWER 1

This is a well-organized manuscript and written in clear language, but following issues should be addressed to improve the content:

  1. In the introduction section, the authors discuss the structure of this manuscript, which is very clear. However, I don’t see much discussion about previous theoretical and numerical studies, and how this work is different from them. The motivation behind this work was not very clear. Thus, I would suggest the authors reframe some portions of the introduction section and include more references to highlight the motivation and novelty of the work.

This was added from (line 50 to 57)  

“Therefore, mathematical models were used to analyze the temperature distribution in the cell from this work, as in other works, e.g., [10], which focused on finding errors in the thermal conductivity measurement process, [11], which studied the impact of the geometry for heat transfer in nanofluids, and [12], which studied a Spray Fluidized Bed Granulator. However, from the literature, no works related to the analysis of the thermal performance to select the geometry of a cell in an adiabatic calorimeter have been found. Attempts to learn the temperature distribution within the cell, the location of the temperature sensor, and the fluid inlets and outlets have been made [13].”

  1. In the caption of Fig.3, c) and d) seem to be the same; however, the corresponding figures are different. Please revise them. Also, the dimensions of the cylinder or sphere of each case can be added to the figure to make it easier for readers.

Thank you for your feedback. It was changed by another plot to avoid confusion.

  1.     Does the numerical model in Fig. 2 shows the one for the cylindrical model? Is it possible to show the spherical model as well?

Thank you for your observation. Because the study of our work focused on the theoretical solution of a cylinder, it is not necessary to add the model of the sphere, in addition to the fact that the sphere only has the radius as a variable.

Reviewer 2 Report

The English in this paper needs a lot of work.  There are subject verb disagreements as well as a number of sentence fragments.  I got the drift of most of what was said, but the massacre of the English language makes me unsure that I did understand what was done. 

The authors are trying to compare two numerical methods without showing either is converged.  This does not make sense.  MATLAB is a numerical procedure not an analytic one.  The finite element method has been validated many times, and I personally would be more likely to expect it to be the more accurate answer rather than the MATLAB approach, if the FE method is converged.

I truly couldn't figure out how the work presented, led to most of the conclusions written in the final section.

It would be useful to get a good English proofreader to correct some of the English mistakes.  

In a number of equations the normalization constants need another parentheses.

There is no reason to assume that the Bessel function/trigonometric function answer is completely correct. Both Bessel functions and trigonometric functions are computed numerically.  Convergence needs to be demonstrated for these as well as the finite element system.

Convergence of the finite element analysis should be demonstrated. Comparing differences without showing both methods are converged, does not provide a valid comparison.

The conclusions are a little fuzzy.  The actual recommendations need to be a little stronger.

Author Response

REVIEWER 2

The English in this paper needs a lot of work.  There are subject verb disagreements as well as a number of sentence fragments.  I got the drift of most of what was said, but the massacre of the English language makes me unsure that I did understand what was done.

The authors are trying to compare two numerical methods without showing either is converged.

Thank you for your feedback. We are comparing the physical phenomenon described in section 2. Theoretical Analysis of two different methods. A theoretical one where the partial differential equation was solved in a transient state that describes the physical phenomenon and another by simulation through the finite-element method with the ANSYS software. For the analytical solution, the values that were taken for p and m in the summations and achieve convergence are shown in line 128. For the numerical solution using the software, the default value was taken of 1X10^-4. Our monitor to assess the convergence of both solutions was the value of the temperature at the border.

This does not make sense.  MATLAB is a numerical procedure not an analytic one.  The finite element method has been validated many times, and I personally would be more likely to expect it to be the more accurate answer rather than the MATLAB approach, if the FE method is converged. 

Thank you for your comment. We used MATLAB to program Equation 30, this is the solution theoretical of the differential partial equation, equation 3 in the paper, therefore more accurate than any numerical method.

I truly couldn't figure out how the work presented, led to most of the conclusions written in the final section. 

We appreciate your comment. The conclusions were obtained from the results presented in the graphs of Figure 3 and throughout the work. However, it was reviewed in the latest version.

It would be useful to get a good English proofreader to correct some of the English mistakes. 

Thank you for your appreciable recommendation. We used in the latest version of the manuscript the English service of MDPI.

In a number of equations the normalization constants need another parentheses. 

Thank you for your kindly observation. We corrected equation 19.

There is no reason to assume that the Bessel function/trigonometric function answer is completely correct.

Thank you for your comment. The equation 3, solved in our work, is supported by [20] Özisik, N.; Bayazitoglu, Y.a. Elements of heat transfer; McGraw-Hill New York etc., 1988. Included in the references section.

Bth Bessel functions and trigonometric functions are computed numerically. 

Thank you for your kindly observation. MATLAB has a mathematical toolkit very extensive, it includes Bessel Functions. Only Equation 26 was programmed because MATLAB has not derivatives of Bessel Functions.

Convergence needs to be demonstrated for these as well as the finite element system. Convergence of the finite element analysis should be demonstrated. Comparing differences without showing both methods are converged, does not provide a valid comparison.

Thank you for your kind suggestions. Let you comment, if the analytical and numerical solutions not converged, all the system would not reach the temperature set at the boundary conditions, represented by Tc. If we observe Fig. 3, we can see that the temperature at the borders is reached.

Line 128 describes the number of iterations for the summations which convergence is reached for the analytical solution. 

For the numerical solution, the default convergence criterion established in ANSYS was used.

The conclusions are a little fuzzy.  The actual recommendations need to be a little stronger.

Thank you for your recommendation. Conclusions were rechecked to improve this section.

Reviewer 3 Report

1-  English and writing should be double-checked. Typos should be removed from the paper.

2-  The topic of calorimetry and thermal engineering is one of the fascinating research areas especially in modeling and experimental studies. Calorimetry and thermal modeling of particulate flows are by far one of the most important research topics, which authors should review in the literature. Adiabatic calorimetry is one of the major approaches for measuring the physical properties of working fluids such as nano-suspensions. Searching the literature, the author can mention the application of calorimetry in thermal engineering. Following papers are suggested to be read and used:

Numerical simulation of natural convection heat transfer of nanofluid with Cu, MWCNT, and Al2O3 nanoparticles in a cavity with different aspect ratios.  Diurnal thermal evaluation of an evacuated tube solar collector (ETSC) charged with graphene nanoplatelets-methanol nano-suspension.  Reforming of methanol with steam in a micro-reactor with Cu–SiO2 porous catalyst

3-  Equation 2 is the Joule’s heating equation, which for AC power has a term COS <Phi>. Phi is the angle between the current and voltage vectors. Hence, COS phi=-1. The authors should amend the equation considering this physical concept.

4-  How accurate is the analytical solution when comparing to the experimental data available in the literature? The authors should elaborate on the text and discuss the accuracy.

5-  Discuss the mesh quality, orthogonality, and also the size of the cells in the mesh structure. It is critical for the authors and helps to replicate the problem.

6-  There are small fluctuations seen in Fig. 3. The authors can elaborate on the reason in the paper.

7-  It is essential to add a one or two paragraph(s) to the paper and discuss the quantitative results obtained in the paper. The conclusion should also show the main contribution of the paper to the existing literature.

8-  Make the gap and objective of the paper clearer in the introduction part. 

Author Response

REVIEWER 3

1-           English and writing should be double-checked. Typos should be removed from the paper.

Thank you for your appreciable recommendation. We used in the latest version of the manuscript the English service of MDPI.

2-          The topic of calorimetry and thermal engineering is one of the fascinating research areas especially in modeling and experimental studies. Calorimetry and thermal modeling of particulate flows are by far one of the most important research topics, which authors should review in the literature. Adiabatic calorimetry is one of the major approaches for measuring the physical properties of working fluids such as nano-suspensions. Searching the literature, the author can mention the application of calorimetry in thermal engineering. Following papers are suggested to be read and used:

Numerical simulation of natural convection heat transfer of nanofluid with Cu, MWCNT, and Al2O3 nanoparticles in a cavity with different aspect ratios.

Thanks for your valuable contribution. These references and others were added in the following lines:

(Linea 19 to 21) Among these substances is nanofluid, which has been recently introduced and improves heat transfer in geometries such as enclosures, channels, and  microchannels (Hoseinzadeh, 2019).

(line 152) Thermal conductivity essential variable in the model (Sarafraz, M. 2019).

(Line 28) In (Sarafraz, M. 2019), heat capacity is an important variable to calculate the thermal efficiency in any system which involves heat transfer.

 (Line 248) Therefore, geometry affects thermal behavior, as reported by (M.M. Sarafraz, 2019)

3-  Equation 2 is the Joule’s heating equation, which for AC power has a term COS <Phi>. Phi is the angle between the current and voltage vectors. Hence, COS phi=-1. The authors should amend the equation considering this physical concept.

Thanks for your feedback. This equation is because in the experimental we will use a power supply DC, in the experimental part not use CA. Besides this equation was obtained from [7] Tan, Z.C.; Shi, Q.; Liu, X. Construction of High-Precision Adiabatic Calorimeter and Thermodynamic. Study on Functional Materials. Calorimetry: Design, Theory and Applications in Porous Solids 2018, p. 1. 232 doi:10.5772/intechopen.76151. Already referenced in our work.

 4- How accurate is the analytical solution when comparing to the experimental data available in the literature? The authors should elaborate on the text and discuss the accuracy.

Thanks for your valuable contribution. (line 211 )Taking account literature (Sun, M.T.; Chang, C.H. The error analysis of a steady-state thermal conductivity measurement method with single constant temperature region. Journal of Heat Transfer], Volumen= 129, Number=9, year=2007) the analytical solution developed in this work is at the most 3% difference for experimental results that could be obtained. Actually we have performed a few experiments and find out a maximum of 3.4% and a minimum for 0.1% at the sphere. For cylinder the minimum  0.1% and a maximum of 4.7%. 

5-       Discuss the mesh quality, orthogonality, and also the size of the cells in the mesh structure. It is critical for the authors and helps to replicate the problem.

(Line 144 to ) Taking into account the numerical solutions, the grid number was investigated as aparameter that affects the accuracy of the obtained results. Based on the variations of these parameters in a grid number of 5610 nodes, we can be assured that the obtained results are accurate, because several simulations were carried out varying mesh size and time step for convergence until get the minimum percentage which was 0.075 % error against the theoretical solution. The surface was meshed using a quadrilateral dominant method as displayed in Figure 2.

6-          There are small fluctuations seen in Fig. 3. The authors can elaborate on the reason in the paper.

Thanks for your valuable observation. Fluctuations are due to data, because the information was taken point-to-point basis and was plotted from the dataset. But these fluctuations are not due to the effects of the variables or the programming obtained in the analytical solution and by simulation.

7-          It is essential to add a one or two paragraph(s) to the paper and discuss the quantitative results obtained in the paper. The conclusion should also show the main contribution of the paper to the existing literature.

Thanks for your valuable contribution. In the latest version of the paper we added:

(line 252 to 256) When heat is added to a system, increasing the temperature from 23 ° C to 25 ° C, the difference in temperature between the analytical solution and the numerical solution is 0 as seen in Figure 3a.

(line 237 to 239)The maximum temperature variation between the cylinder and the sphere obtained from the simulation, occurred at the time of extracting heat from the system, causing a decrease in temperature from 30 ° C to 29 ° C, generating a temperature variation of 0.08 ° C according to Figure 3e.

(line 245 to 251) The spherical geometry have better thermal performance than cylindrical, because the gradients temperature are smaller, from results obtained was found that the maximum gradient for spherical cell is 0.104◦C, and the maximum gradient for the cylindrical cell is 0.208◦C therefore, geometry affects thermal behavior, as reported by [2]. In real applications, the system in a sphere responds with greater thermal speed than in a system contained within a cylinder. But the homogeneity in the fluid contained in the cell is the most important variable because of affect accuracy measurements in heat capacity value.

8-          Make the gap and objective of the paper clearer in the introduction part

(line 75 to 82) The main objective of this work was to select the geometry with the lowest temperature gradients through carrying out an analytical solution of the thermal behaviour of the cell from an adiabatic calorimeter to later compare it with a numerical solution, and to determine the error between both methods to provide greater reliability in the simulation of processes, where there is an exchange of energy with geometries of greater complexity, allowing for a smaller number of experimental prototypes that will reduce time and cost in the design of any thermal system. Additionally, with this information, finding the best thermal performance between a cylindrical vs. spherical geometry for the development of the cell of an adiabatic calorimeter.

Reviewer 4 Report

From the title of the paper one can conclude that the authors proposed a new numerical approach to the analysis of heat transfer in adiabatic calorimeter that can improve quantitative analysis of the results obtained in that type of calorimeters. Unfortunately, this is not the case. The paper contains two main parts. In the first one an analytical solution of the problem of heat transfer in a cylinder is presented with details. As can be concluded (line 106) it was taken from the literature (Ozisik; with mistakes in References). Th second part is on the numerical solution of the same problem. However, the objective of the analysis is extremely simplified in terms of geometry and boundary conditions and it does not reflect the real conditions in calorimeters.

Selected remarks.

First, general remark – the language of the paper is far from the level which is required in academic publications. For example, the description of the operation of calorimeters, which is given in the beginning of Introduction, is completely unclear for those who are unfamiliar with the topic. In the rest of the paper there are many incorrect and unclear wordings.

Other:

Reference to eq. (3) in line 36 incorrect.

I am not sure whether references [12 and 13] are appropriate for the analytical solution of the problem presented in section 2.1. The same for references [21, 23, 24, 27].

A formula in line 84 looks a little strange.

An assumption for thermal diffusivity (in lines 123-124) is unclear.

How was the “error percentage” defined (line 128). The dependency of an error on temperature range (observed in the study) results mainly from its definition.

Fig.3: Comparison of temporal temperature variations in objects of different shapes (cylinder and sphere) is not a good idea, what is a goal of such a comparison?

Lines 152-153: “Matlab is a high-level tool for developing technical applications”; Matlab is not a tool for developing technical applications.

The largest paragraph in the part Discussion is on the benefits of Matlab.

The first statement in Conclusions: “A strategy was established to improve computational time”, however, I couldn’t find any information on such strategy related to the operation of adiabatic calorimeter, which should be the main goal of the research.

Author Response

REVIEWER 4

From the title of the paper one can conclude that the authors proposed a new numerical approach to the analysis of heat transfer in adiabatic calorimeter that can improve quantitative analysis of the results obtained in that type of calorimeters. Unfortunately, this is not the case.

Thank you for your recommendation. The title was changed by

Finite-Element Simulation for Thermal Modeling of a Cell in an Adiabatic Calorimeter

The paper contains two main parts. In the first one an analytical solution of the problem of heat transfer in a cylinder is presented with details. As can be concluded (line 106) it was taken from the literature (Ozisik; with mistakes in References).

Thanks for your valuable contribution. The reference was checked and corrected. 

General solutions were taken from  [20] Özisik, N.; Bayazitoglu, Y.a. Elements of heat transfer; McGraw-Hill New York etc., 1988. but the development of the complete analytical solution was made by the authors. Then, this solution is not reported in any book or article.

The second part is on the numerical solution of the same problem. However, the objective of the analysis is extremely simplified in terms of geometry and boundary conditions and it does not reflect the real conditions in calorimeters.

Thanks for your feedback. This is true, the problem was simplified. However, it is established in the paper that only the behavior of the cell, like a part of the calorimeter and not the entire calorimeter was analyzed. So in line 141 to 142 is mentioned that the model was analysed in 2D dimension.

Selected remarks.

First, general remark – the language of the paper is far from the level which is required in academic publications. For example, the description of the operation of calorimeters, which is given in the beginning of Introduction, is completely unclear for those who are unfamiliar with the topic. In the rest of the paper there are many incorrect and unclear wordings.

Thank you for your appreciable recommendations.  We used in the latest version of the manuscript the English service of MDPI. 

Other:

Reference to eq. (3) in line 36 incorrect.  

Thank you for your observation. It was corrected.

I am not sure whether references [12 and 13] are appropriate for the analytical solution of the problem presented in section 2.1. corregido es cierto

The same for references [21, 23, 24, 27]

Thank you for your suggestion. It was analysed and they were removed, [12,13 y 27].

For [23 and 24] was located in the adequate paragraph.

A formula in line 84 looks a little strange. 

The formula was complemented. Currently it is in (Line 118)

An assumption for thermal diffusivity (in lines 123-124) is unclear. 

 The explanation was complemented by line 165 and 166, additional complement is in line 233 to 236

How was the “error percentage” defined (line 128). The dependency of an error on temperature range (observed in the study) results mainly from its definition.  

The error was calculated taking the value obtained in the analytical solution as the true value. So, the difference between the absolute error with respect to the true value divided by  the true value.

Fig. 3: Comparison of temporal temperature variations in objects of different shapes (cylinder and sphere) is not a good idea, what is a goal of such a comparison? 

We appreciate the observation. One of the goals is to know the speed of the response to a change in temperature, because the operation of the calorimeter will be in a transitional state.

Lines 152-153: “Matlab is a high-level tool for developing technical applications”; Matlab is not a tool for developing technical applications.

Thank you for your kindly comment. It was changed by “Matlab is a high-level tool for development of applications in engineering and research”. in line (199).

The largest paragraph in the part Discussion is on the benefits of Matlab. 

Thank you for your support. The paragraph was modified, nad is showed in 199 to 202.

The first statement in Conclusions: “A strategy was established to improve computational time”, however, I couldn’t find any information on such strategy related to the operation of adiabatic calorimeter, which should be the main goal of the research.

In this sentence we are referring to the computation time and data handling in the analytical and numerical solutions, but not to the operation of adiabatic calorimeters. So, Figure 2a, the physical properties of water were used, there is an answer in 3000 seconds, unlike the other graphs where the answer is the same but in less time.

Reviewer 5 Report

General comments :

In this study, authors introduce the development and the verification/validation of a their adiabatic calorimeter by the solution of thermal conduction. They established: (1) a theoretical and numerical FEM model for cylindrical calorimeter cell; (2) a numerical FEM model for spherical calorimeter cell. The manuscript is nicely conceived in a logical order. The subject matter is appropriate for Energies readers. The manuscript needs a thorough English check. Discussion points out nice perspectives but the conclusions are mostly qualitative. So, improvements are needed on the description of methods and on the discussions by providing quantitative evidence.

Detailed comments :

  • Lines 36-37: Equation (2). Effect of voltage not clear. Voltage or R ? Check ref [8].
  • Lines 45-48: Why the theoretical solution is provided only for cylindrical cell ? Is it only a benchmarking of the FEM model ?
  • Line 54: The theoretical model is written in Matlab, so the analytical solution was somehow transferred into numerical form. So, are there any computational errors expected, in the analytical model, due to this numerical transcription ?
  • Line 57: Could you kindly indicate the actual software used under Ansys platform? (Thermal ? Structural Thermal ?) . Is it a FEM or FVM ? I assume it is FE (line 109: “8-node element”). Please kindly indicate all the software versions, the CPU runtimes and the computational configurations used. Please name if any special toolbox was used. Those are important for the repeatability of study.
  • Line 66: I assume you mean “spacial”.
  • Line 98: In authors’ opinion, what are the advantages/disadvantages of analytical solution compared to numerical (finite element code), because today’s softwares are quite efficient in terms of computational time & memory consumption ? Could you provide a brief discussion to highlight the added value of the present study ?
  • Line 114: What is the timestep for convergence and the actual physical timestep for transient solution ? As authors may have noticed, solution of conduction problem is governed by the characteristic time of heat conduction/diffusion, while the mesh size is less important.
  • Figures 3: Is the convergence checked, the mesh convergence and timestep convergence, before plotting the comparative T° distributions ? Please indicate.
  • Water is considered perfectly stationary, any convection ignored. Is the assumption valid and why? Please explain.
  • Line 151: “high temperature performance”, for e.g. which is the high temperature range according to authors ?
  • Line 165-168: Very nice approach. Could you please give quantitative results ? Regarding the conduction and heat leaks, for example, what are the optimal conditions of heat ramps/loading in adiabatic calorimeter, as a function of materials conductivity/diffusivity ? This would be a very interesting result to indicate; and it shows the added value of the study.
  • According to authors, an added value of study is the rapidity of analytical solution. How fast is the analytical solution compared to 3D simulation, any quantiative indicator ?
  • Lines 184-187: Not clear, please kindly revise the sentences.
  • Lines 199-208: It is not clear why further literature references are provided in conclusion. Please consider moving them in the literature review, in introduction.

Author Response

REVIEWER 5

General comments :

In this study, authors introduce the development and the verification/validation of a their adiabatic calorimeter by the solution of thermal conduction. They established: (1) a theoretical and numerical FEM model for cylindrical calorimeter cell; (2) a numerical FEM model for spherical calorimeter cell. The manuscript is nicely conceived in a logical order. The subject matter is appropriate for Energies readers. The manuscript needs a thorough English check. Discussion points out nice perspectives but the conclusions are mostly qualitative. So, improvements are needed on the description of methods and on the discussions by providing quantitative evidence.

Thank you for your appreciable recommendation. We used in the latest version of the manuscript the English service of MDPI. It was improved the introduction and conclusion parts.

Detailed comments :

  •       Lines 36-37: Equation (2). Effect of voltage not clear. Voltage or R ? Check ref [8]. 

There is a mistake with the reference it was changed by: Tan, Z.C.; Shi, Q.; Liu, X. Construction of High-Precision Adiabatic Calorimeter and Thermodynamic 231 Study on Functional Materials. Calorimetry: Design, Theory and Applications in Porous Solids 2018, p. 1. doi:10.5772/intechopen.76151.

And was definen in the paper such as  and the heat added is obtained by the electrical energy calculated by Equation 2... in the line 42

  •       Lines 45-48: Why the theoretical solution is provided only for cylindrical cell ? Is it only a benchmarking of the FEM model ?

Thank you for your recommendation. We tried to solve theoretically for the spherical one. However, it was not achieved. Therefore, it was decided to only do the analysis numerically. Also, the cylindrical cell are the most used. 

  •       Line 54: The theoretical model is written in Matlab, so the analytical solution was somehow transferred into numerical form. So, are there any computational errors expected, in the analytical model, due to this numerical transcription ? 

The only possible error added to the numerical solution can be by the defined summations shown in equation 30, these was adjusted by varying the number of p and m, and then the 15 decimal places after the point that MATLAB shows in the screen is 9, enough to consider it exact. i.e. if the border temperature (Tc) established was 30, the result obtained was 29.999999999999999.

  •       Line 57: Could you kindly indicate the actual software used under Ansys platform? (Thermal ? Structural Thermal ?) . Is it a FEM or FVM ? I assume it is FE (line 109: “8-node element”). Please kindly indicate all the software versions, the CPU runtimes and the computational configurations used. Please name if any special toolbox was used. Those are important for the repeatability of study.

Thank you. The requested information was added to line 137 and 140

(line 137 to 140) It is used the Ansys Mechanical Parametric Design Language (APDL) user interface 18.0, which uses the finite-element method to approximate the solution of transient heat transfer conduction by Equation (4) show above [13,14] for the numerical solution running in a workstation with Xeon processor of 16 cores and 16 GB in RAM memory.

(line 155) The study was solved in transient mode for a time of 6000s with a time step size of 0.1.

  •       Line 66: I assume you mean “spacial”

Thank you for your observation. It was changed.

  •       Line 98: In authors’ opinion, what are the advantages/disadvantages of analytical solution compared to numerical (finite element code), because today’s softwares are quite efficient in terms of computational time & memory consumption ? Could you provide a brief discussion to highlight the added value of the present study ?

Thank you for your recommendation. It was added to line (line 75 to 82) The main objective of this work was to select the geometry with the lowest temperature gradients through carrying out an analytical solution of the thermal behaviour of the cell from an adiabatic calorimeter to later compare it with a numerical solution, and to determine the error between both methods to provide greater reliability in the simulation of processes, where there is an exchange of energy with geometries of greater complexity, allowing for a smaller number of experimental prototypes that will reduce time and cost in the design of any thermal system. Additionally, with this information, finding the best thermal performance between a cylindrical vs. spherical geometry for the development of the cell of an adiabatic calorimeter.

Line 114: What is the timestep for convergence and the actual physical timestep for transient solution ?As authors may have noticed, solution of conduction problem is governed by the characteristic time of heat conduction/diffusion, while the mesh size is less important. 

Thank you for your appreciable recommendation. It was added to line 155.

  •       Figures 3: Is the convergence checked, the mesh convergence and timestep convergence, before plotting the comparative T° distributions ? Please indicate.

(Line 147) ...because several simulations were carried out varying mesh size and time step for convergence until get the minimum percentage which was 0.075%  error against the theoretical solution.

  •       Water is considered perfectly stationary, any convection ignored. Is the assumption valid and why? Please explain.

Because of the proposed system and the evaluations carried out with the established boundary conditions, the temperature variation is so small that a phase change is not reached that generates convective effects in the water.

  •       Line 151: “high temperature performance”, for e.g. which is the high temperature range according to authors

Thank you. This line was removed due to improvements in the wording. However, they were referring to temperatures reported by Magee, from 420 to 720 K.

  •       Line 165-168: Very nice approach. Could you please give quantitative results ? Regarding the conduction and heat leaks, for example, what are the optimal conditions of heat ramps/loading in adiabatic calorimeter, as a function of materials conductivity/diffusivity ? This would be a very interesting result to indicate; and it shows the added value of the study

At the moment we have not carried out experimentation to measure heat capacity. However, values on the maximum and minimum gradients were added for the selection of the most convenient geometry on lines 245 to 248.

  •       According to authors, an added value of study is the rapidity of analytical solution. How fast is the analytical solution compared to 3D simulation, any quantiative indicator ?

An approximate value is that for the analytical solution programmed in MATLAB it takes around 30 seconds and for the numerical solution using ANSYS approximately 90 seconds. This information is not relevant because, as the model is simple, it does not require long simulation times on the order of time or days.

  •       Lines 184-187: Not clear, please kindly revise the sentences.

Thank you for your support. Lines have been checked.

  •       Lines 199-208: It is not clear why further literature references are provided in conclusion. Please consider moving them in the literature review, in introduction.

Thank you for your support. Lines have been checked.

Round 2

Reviewer 2 Report

  1. Words in lines 81 to 83 are not a sentence but a sentence fragment.  Suggest you get a better English proofreader than MDPI
  2. The Normalization constants in equations 23 and 29 appear to have tha parentheses in the wrong place.
  3. Please reference the solution convergence criteria that led to 65 values for pmax and 200 values for mmax.

Author Response

  1. Words in lines 81 to 83 are not a sentence but a sentence fragment.  Suggest you get a better English proofreader than MDPI. Ans: The sentence has been rewritten to: "Also, the calculated information given by the simulation contributes to a criteria design. It permits to choose a spherical geometry instead of a cylindrical, as reported in the state-of-the-art.  ". Besides, language has been rechecked.
  2. The Normalization constants in equations 23 and 29 appear to have the parentheses in the wrong place. Ans: We appreciate your kind comment. Thus, we adjusted the parentheses the equations.
  3. Please reference the solution convergence criteria that led to 65 values for pmax and 200 values for max. Ans. We appreciate your kindly observation. We calculated Figure 3 (a) by considering pmax and mmax values, the analytical solution gives 24.999. Then, when increasing those values from 65 to 70 and 200 to 250 or even higher, the results remain constant in 0.9. If those values are decreased to less than 65 and 200, the analytical solution gives to 24.8999. Therefore, those values were chosen because the error is the minimum.

Reviewer 3 Report

Accept as is

Author Response

Thank you for your advices 

Reviewer 4 Report

The paper has been significantly improved, now it looks much better.

Author Response

Thank you for your feedback to improve the manuscript.

Reviewer 5 Report

Overall manuscript is improved. Minor spell checks shall be needed.

Author Response

Thank you for your kind advice. The manuscript was checked by a native speaker. Moreover the MDPI English service (English-18127). Also, we recheck the spelling of the paper.